# Module-Fluidics: Building Blocks for Spatio-Temporal Microenvironment Control

**DOI:** 10.3390/mi13050774

**Published:** 2022-05-14

**Authors:** Bowen Ling, Ilenia Battiato

**Affiliations:** Energy Resource Engineering, Stanford University, Stanford, CA 94305, USA; lingbowen@imech.ac.cn

**Keywords:** mircofluidics, micromodel, microfluidic valve

## Abstract

Generating the desired solute concentration signal in micro-environments is vital to many applications ranging from micromixing to analyzing cellular response to a dynamic microenvironment. We propose a new modular design to generate targeted temporally varying concentration signals in microfluidic systems while minimizing perturbations to the flow field. The modularized design, here referred to as module-fluidics, similar in principle to interlocking toy bricks, is constructed from a combination of two building blocks and allows one to achieve versatility and flexibility in dynamically controlling input concentration. The building blocks are an oscillator and an integrator, and their combination enables the creation of controlled and complex concentration signals, with different user-defined time-scales. We show two basic connection patterns, in-series and in-parallel, to test the generation, integration, sampling and superposition of temporally-varying signals. All such signals can be fully characterized by analytic functions, in analogy with electric circuits, and allow one to perform design and optimization before fabrication. Such modularization offers a versatile and promising platform that allows one to create highly customizable time-dependent concentration inputs which can be targeted to the specific application of interest.

## 1. Introduction

Microfluidic systems are key tools routinely used in biological, chemical, environmental and materials engineering and science since they allow one to realize controllable micro-environments [1]. Microfluidics systems fabricated by soft-lithography technique are low cost, microbially compatible and highly resistant to various chemicals [2], which render them a default choice in many fields [3,4,5,6,7,8,9,10]. Some applications include cell stimulation [11], single-molecule monitoring [12], micro-scale mixing and concentration control [13,14,15], micro synthesis [16] and solid-free gel-casting [17]. The design objectives of a specific microfluidic system may be extremely different, ranging from enhancing reactions through mixing [18,19] to creating a dynamic solute concentration (in mol/L) profile for cell stimulation [20,21]. Yet, the underlying principle is to control concentration gradients ∇c ( concentration difference of a unit length, mol/L·m) in the device through *ad hoc* reconfiguration of the underlying flow field. This can be achieved by both passive and active control mechanisms. Passive control is based on the principle of restructuring the flow to improve, e.g., mixing efficiency, through appropriately designed geometrical features in the chip, such as channel lengths and/or micropatterns that introduce perturbations in the flow field [22,23]. Instead, active control uses mechanical (e.g., pressure, sound wave) or non-mechanical (e.g., electrical) exterior forces to influence the flow field within the device [7]. The primary disadvantage of relying on flow field disturbances to control transport (i.e., ∇c) is a fundamental lack of flexibility in the design, particularly when specific experimental conditions, different from those the device has been originally designed for, are desired. For example, since passive control relies on geometrical features of the system, which are hard-wired to a chip, it is challenging to produce different solute/mixing conditions in the same device [11,22,23]. In systems employing active control, which provides more flexibility in terms of controlling the system state, ∇c and viscous dissipation, i.e., energy consumption, are always interlinked. As a result, active control based on flow reconfiguration may be limited by the application (e.g., with living cells) or the yielding stress of the device [3]. There are only a few control mechanisms that can generate a steady flow field while manipulating the concentration field [11]. Yet, most of these designs heavily rely on the specific geometry of the device itself, and lack flexibility for various applications’ needs.

In all such systems, the control ability is constrained by the flow and size of the microfluidic device. Furthermore, in classical designs of PDMS multi-layered microfluidic systems, valves are hard-wired to the experimental chip: as a result, testing and optimization lead to re-designing and re-fabrication [24], and the process requires not only specialists’ input but also specific facilities and equipments. This fundamentally limits the application of the technology in non-scientific communities [1].

We propose a novel and flexible design, here referred to as Module-Fluidic, to directly control ∇c by creating temporally varying input concentration signals while keeping the flow steady. This is achieved through (i) a system of microfluidic valves that controls the inlet concentration and (ii) a modularized design, separate from the experimental chip, which includes two modules (an oscillator and an integrator) to generate complex input signals. Specifically, in the proposed design none of the components is hardwired to the experimental chip where tests need to be conducted (e.g., mixing, cell sorting etc.), and individual modules can be recombined to generate different input signals on the same experimental chip. The importance of modularization, shown in Figure 1, is three-fold: it allows one: (i) to isolate the input signal generation system, composed of moving parts, from the experimental chip itself so that modifications of the input conditions do not require any redesign and refabrication of the experimental chip; (ii) to create a library or menu of modules of basic signals; and (iii) to use predesigned modules to generate complex signals by standard in-series and in-parallel connections in analogy with electric circuits. This new design enables a precise control of the injected solute concentration with a steady flow field, this condition is relevant for minimizing disturbance and enhancing repeatability for benchmark experiments using microfluidics [4,25].

## 2. Materials and Methods

### 2.1. Signal Generator Design: Oscillator

The key of the proposed design is to provide a dynamic concentration input while the flow field is kept steady. To achieve this goal, we combine two symmetric flow paths to generate temporal variations in concentration. The design includes four microfluidic valves V${}_{I}$, V${}_{II}$, V${}_{III}$ and V${}_{IV}$, as shown in Figure 2A,B. Each valve is a dead-end channel that lies beneath the flow path; the overlapping area between the flow path and the valve is a thin PDMS membrane. When a pressure (∼20 psi) is applied to the valve, the membrane balloons toward the flow layer and seals the fluid path. To ensure a complete seal, the flow channel must be semi-cylindrical, and a positive resistance is used to achieve the desired cross-section. All the valves are controlled by a microfluidic valve control matrix (MUX QUAKE VALVE, Elveflow) with sixteen independent pressure outlets that can provide a constant pressure input when they are switched on. The control matrix can be programmed by a PC. The programmable valve control matrix can keep each valve open or closed for the desired time: this allows one to generate different input functions. Two inlets ($I_{I}$ and $I_{II}$) are connected to the same pump that provides a constant flow rate during the experiment. Inlets $I_{I}$ and $I_{II}$ are connected to DI-water (C=0) and concentrated solution (C=1) syringes, respectively. By alternating the valves on-off combination, the switch between the two injections ports occurs while keeping the flow steady. If the valves $V_{I}$ and $V_{III}$ are closed, and $V_{II}$ and $V_{IV}$ are open, the concentrated solution (C=1) enters the experimental chip through $I_{II}$ while the DI-water solution is diverted toward the exit. The opposite set of valves allows one to obtain a C=0 signal instead, see Figure 2C. The signal generated by the oscillator is stable and can be analytically represented by the function:(1)$$\begin{array}{c} C_{\mathrm{oscillator}}^{\mathrm{out}}\left(t\right)=\left\{\begin{array}{cc} \sin\left(\frac{2\pi t}{\omega }\right), & t\in \left(nT_{p},nT_{p}+\frac{\omega }{4}\right)\\ 1, & t\in \left(nT_{p}+\frac{\omega }{4},nT_{p}+\frac{\omega }{4}+T_{C1}\right)\\ \sin\left[\frac{2\pi (t-T_{C1}-\frac{\omega }{4})}{\omega }\right], & t\in \left(nT_{p}+\frac{\omega }{4}+T_{C1},nT_{p}+\frac{\omega }{2}+T_{C1}\right)\\ 0, & t\in \left(nT_{p}+\frac{\omega }{2}+T_{C1},\left(n+1\right)T_{p}\right), \end{array}\right. \end{array}$$
where
(2)$$\begin{array}{c} T_{p}=\frac{\omega }{2}+T_{C1}+T_{C2},\;n=\left\{1,2,3,\cdots \right\}, \end{array}$$
and the time scales ω, $T_{C1}$, and $T_{C2}$, defined in Figure 2D, are related to the valve response time (ω/4), and the lengths of the C=1 and C=0 injection phases, respectively.

In Figure 3A, we show two measured concentration signals with different characteristic time scales, and the fitted analytic function (Equation 1) with $T_{C1}=T_{C2}=0.61$ s, ω=1.5 s and $T_{p}=1.97$ s, see Figure 3A (top), and $T_{C1}=0.63$ s, $T_{C2}=2.65$ s, ω=1.5 s and $T_{p}=4.03$ s, see Figure 3A (bottom).

### 2.2. Modularization: Integrator

Another key feature of the proposed design is modularization, i.e., the ability to combine predesigned modules from a library or menu of basic signals to generate complex ones. Yet, the vastly different characteristic volumes between piping systems (mL scale) and the microfluidic signal generator ($10^{-6}$–$10^{-3}$ mL) prevents the use of standard tubes and connectors to combine oscillators with different characteristic time scales. To overcome this issue, we use a microfluidic channel itself as the connector between modules and we refer to it as an ‘integrator’ due to its dispersive effects on the concentration signal.

The integrator is a single microfluidic channel with two pins. Due to dispersive effects within the pins and the channel, the connector acts as an integrator with a fixed integration window $T_{o}$, i.e., it averages the concentration input signal according to the analytical expression:(3)$$C_{\mathrm{integrator}}^{\mathrm{out}}\left(t\right)=\frac{1}{T_{o}}\int_{t-T_{o}}^{t}C^{\mathrm{in}}\left(t^{*}\right)dt^{*},$$
where $C^{\mathrm{in}}$ is the input concentration signal and $C_{\mathrm{integrator}}^{\mathrm{out}}$ is the integrator output signal. The characteristic time window $T_{o}$ represents the total time required for the fluid to travel from the inlet to the outlet of the integrator. For the specific device we have fabricated (including two pins and the channel), $T_{o}=43.2$ s. Figure 3B shows the output signals of the integrator, $C_{\mathrm{integrator}}^{\mathrm{out}}$, for two input signals with $T_{C1}=T_{C2}=30$ s, ω=1.5 s and $T_{p}=60.75$ s in Figure 3B (top), and $T_{C1}=T_{C2}=55$ s, ω=1.5 s and $T_{p}=110.75$ s in Figure 3B (bottom). The integrator performs an integration operation on the input signal and generates a triangular wave signal as output. The experimentally measured output signals match with the analytical expression (Equation 3) (solid lines in Figure 3B). Additionally, if $T_{p}/2<T_{o}$, i.e., the integration window is larger than the half period of the input signal, then $\min(C_{\mathrm{integrator}}^{\mathrm{out}})>0$ and $\max(C_{\mathrm{integrator}}^{\mathrm{out}})=1$. Instead, when $T_{p}/2>T_{o}$, $\min(C_{\mathrm{integrator}}^{\mathrm{out}})=0$ and $\max(C_{\mathrm{integrator}}^{\mathrm{out}})=1$, i.e., the generated signal spans all the concentration values between 0 and 1.

### 2.3. System Integration: Sampling and Superposition

We propose two basic connections with two oscillators and one integrator: (i) an in-series connection which corresponds to signal sampling and (ii) an in-parallel connection that produces complex superposed signals.

The in-series connection consists of two oscillators and one integrator as represented in Figure 4A (Left). The inlets of the first oscillator are connected to C=0 and C=1, while its outlet is connected, through the integrator module, to one of the inlets of the second oscillator (the other inlet connected to C=0). The resulting signal corresponds to sampling with a given sampling frequency between the input signals C=0 and the signal at the outlet of the integrator. If the signal from the first signal oscillator has characteristic times $T_{C1}$, $T_{C2}$, ω and $T_{p}$, and $T_{o}$ is the characteristic time scale of the integrator, then the outlet signal from a serial connection can be mathematically described as:(4)$$\begin{array}{c} C_{\mathrm{out}}^{--}=\frac{1}{T_{o}}\int_{t-T_{o}}^{t}C\left(t^{*}\right)dt^{*}\cdot \left[H\left(t-t_{1}\right)-H\left(t-t_{2}\right)\right],\;n=1,2,3,\ldots , \end{array}$$
where $C_{\mathrm{out}}^{--}$ is the signal measured at the integrator outlet (whose expression satisfies (Equation 3)), H(·) is the Heaviside function, $t_{1}=nT_{p}+\frac{\omega }{4}$ and $t_{2}=nT_{p}+\frac{\omega }{4}+T_{C1}$. In Figure 4A (Center), we show two experimentally measured concentration profiles (solid lines) generated by a sampling signal with ($T_{C1},T_{C2},\omega ,T_{p})=\left(0.5,0.5,1.5,1.75\right)$ s or ($T_{C1},T_{C2},\omega ,T_{p})=\left(1.5,1.5,1.5,3.75\right)$ s and a signal measured at the inlet of the integrator with ($T_{C1},T_{C2},\omega ,T_{p})=\left(65,65,1.5,130.75\right)$ s or ($T_{C1},T_{C2},\omega ,T_{p})=\left(125,125,1.5,250.75\right)$ s, respectively. Figure 4A (Right) shows a comparison between the experimentally measured signals and Equation (Equation 4). The agreement between the data and the theoretical prediction is good and small discrepancies can be attributed to light intensity variations between different points in the chip.

The parallel connection consists of three oscillators and two integrators, where, however, one of the oscillators is employed exclusively as the concentration measuring point. The parallel connection is shown in Figure 4B (Left). If $C_{1}\left(t\right)$ and $C_{2}\left(t\right)$ represent the output signals of the two upstream integrators and $T_{o}$ their characteristic time scale; the resulting signal is a superposition of the two input signals, i.e.,
(5)$$\begin{array}{c} C_{\mathrm{out}}^{=}=\frac{1}{T_{o}}\int_{t-T_{o}}^{t}C_{1}\left(t^{*}\right)dt^{*}+\frac{1}{T_{o}}\int_{t-T_{o}}^{t}C_{2}\left(t^{*}\right)dt^{*}. \end{array}$$

In Figure 4B (Center), we show the experimentally measured input signals $C_{1}\left(t\right)$ and $C_{2}\left(t\right)$ (in gray), as well as the output function $C_{\mathrm{out}}^{=}$ (colored solid line) for two different cases. Figure 4B (Right) shows a comparison between the experimental measurement and the analytical prediction, Equation (Equation 5). As for the connection in parallel, the agreement between data and model is good. The match between experimental data and the analytically calculated signals allows one to optimize the design for a desired signal generation before any module fabrication.

## 3. Results and Conclusions

We demonstrated that the modulized micorfluidic units provide consistent outcomes as designed, and the combination of units exhibits more complex concentration signal processing capability. From Figure 3 we can see that the oscillator unit generates signals of different intervals steadily, and the signals are highly repeatable as the designed function (Equation 1). Similar result can be seen from Figure 3B, where the integrator processes the input function according to Equation (Equation 3). For the in-series connection, the high frequency result shows a slight deviation at early time, however the overall behavior is stable and it is capable of sampling from the input concentration signal. The parallel connection also fulfills Equation (Equation 5) with a high accuracy.

To conclude, we proposed a novel design to generate time varying concentration signals in microfluidic chips. The design includes (i) a multi-layer microfluidic oscillator, which can produce stable programmable temporal concentration signals and (ii) a microfluidic connector, also referred to as integrator, which enables system modularization through the connection between different components of the microfluidic system, e.g., between different oscillators or between the signal generation system and the experimental chip. Furthermore, the design of two basic connections (in-series and in-parallel), combined with the two basic modules previously designed, can perform signal generation (oscillator), integration (integrator), sampling (serial connection) and superposition (parallel connection). All such signals can be fully characterized by analytic functions, in analogy with electric circuits, and allow one to perform design optimization before fabrication. It is worth emphasizing that modularization allows one to create highly customizable time-dependent concentration inputs which can be targeted to the specific application of interest. Additionally, we plan to explore the applicability of this new design in the following applications: (1) nutrient transport and micro-environment control in micro-bio-reactors, and (2) time-varying solute injection benchmark experiments. Moreover, we aim at extending our design to real application scenarios, where solutions such as cell culture medium, glucose, drug solution and other application-oriented solutions will be tested.

## Figures and Tables

**Figure 1 micromachines-13-00774-f001:**
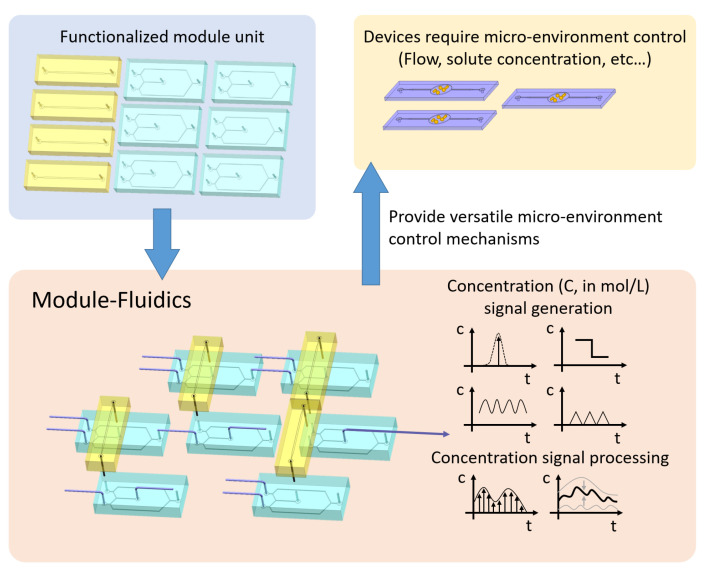
The Module-Fluidic generates the desired concentration profile that enables downstream microfluidic devices’ function.

**Figure 2 micromachines-13-00774-f002:**
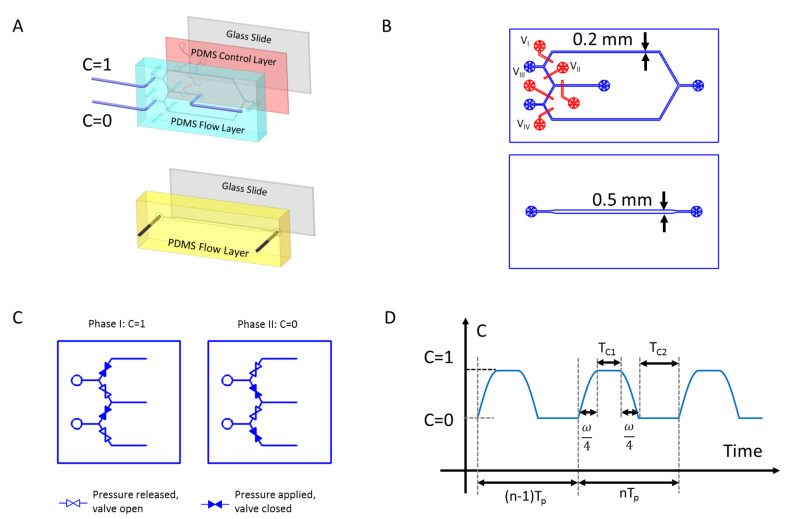
Structures of the Spatio-Temporal Concentration Controller. (**A**) 3D structure of Signal generator (Oscillator) and Module Connector (Integrator), the Oscillator consists of three layers, the PDMS structure layer has flow channels patterned. The channel is molded using positive photo resist which produces a semi-cylindrical cross-section for complete sealing of the micro valve. The second layer is a thin PDMS membrane with controlling valves when the air pressure (∼20 psi) is applied, valve will seal the channel. The third layer is glass slide to support the entire structure. All layers are bounded to each other using plasma bonding. The Integrator is a single microfluidic channel with depth 25 µm. The PDMS layer is also bounded to a glass slide; (**B**) Drawings of the oscillator and integrator (**C**) Concentration signal generation achieved by combination of valve status; (**D**) Function generated by the oscillator.

**Figure 3 micromachines-13-00774-f003:**
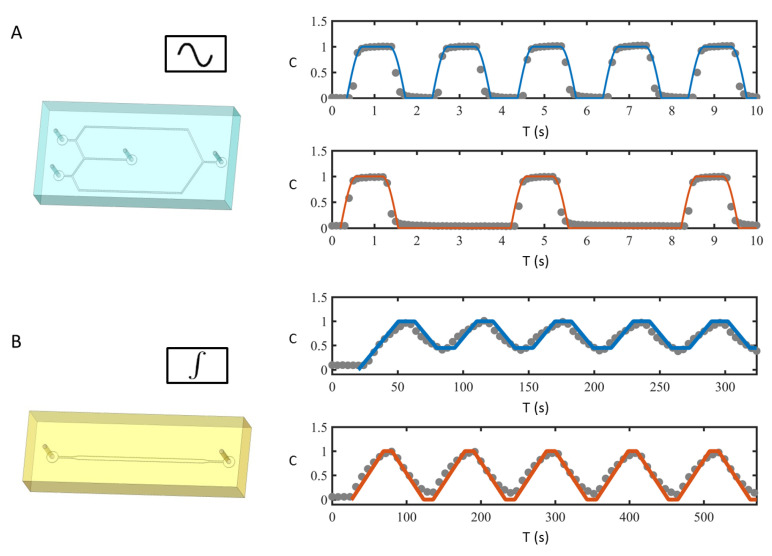
(**A**) (**left**) Oscillator module: (**right**) two signals (symbols) experimentally measured at the outlet of the oscillator and the fitted analytic functions (lines) given by Equation (Equation 1); (**B**) (**left**) Integrator: (**right**) concentration profiles (symbols) measured at the outlet of an integrator connected to an oscillator and the fitted analytic function given by Equation (Equation 3) (lines).

**Figure 4 micromachines-13-00774-f004:**
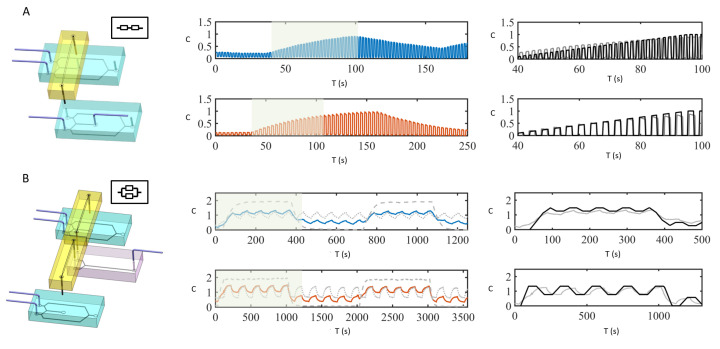
Two basic connection patterns. (**A**) (**Left**) Serial connection: it consists of two oscillators and one integrator, such that the first oscillator provides a slowly varying signal and the second one samples between the input signal and a referencing point (e.g., C=0); (**Center**) The solid lines represent experimentally measured concentration profiles for two different oscillator settings; (**Right**) Comparison between experimentally measured signal (solid gray line) and the analytic prediction in Equation (Equation 4) (solid black line) in the shaded time-window. The analytical solutions are calculated with $T_{C1}=T_{C2}=55$ s for the input function and $T_{C1}=T_{C2}=0.5$ s for the sampling (**top panel**) and $T_{C1}=T_{C2}=55$ s for the input function and $T_{C1}=T_{C2}=1.6$ s for the sampling (**bottom panel**). (**B**) (**Left**) Parallel connection: it contains two integrators and three oscillators, where the two oscillators (top and bottom in the figure) connect directly to the injections and can produce different signals, while the one in the middle is used as a Y-connector and is the measuring point of the outcome concentration. (**Center**) The solid lines represent experimentally measured concentration profiles for two different oscillator settings; (**Right**) Comparison between the experimental signal (solid gray line) and the analytical expression Equation (Equation 5) (solid black line) in the shaded time-window. The analytical solutions are calculated with input signals $T_{C1}=T_{C2}=35$ s and $T_{C1}=T_{C2}=320$ s (**top panel**) and $T_{C1}=T_{C2}=110$ s and $T_{C1}=T_{C2}=1000$ s (**bottom panel**).

## Data Availability

Not applicable.

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
