# Peer review of "Module-Fluidics: Building Blocks for Spatio-Temporal Microenvironment Control"

_micromachines, 2022, doi:10.3390/mi13050774_

Round 1

Reviewer 1 Report

LEGO-fluidics: Building blocks for Spatio-temporal microenvironment control

Although this modular/LEGO-like microfluidic has been published many times and explored extensively, this paper is of good quality. Please do check in with your legal department at Stanford/CAS as the company does not enjoy people using the word "LEGO" without their permission. 

  1. it is rather confusing the way "concentration gradients" are used in this. Concentration gradients mean different things in transport phenomena, biology, or chemistry. This needs to be better defined. C needs to be clearly defined and explained in figure 1. 
  2. the paper could use a little better "outlook": what is the going to be the application uniquely enabled by this function? 
  3. Figure 4 part b you don't necessarily need the third chip in the middle, do you? if you combine the two outputs as an output it would be more accurate than what you illustrate in the small black and white icon at the upper right corner? if so, please delete the third chip and make it a clearer illustration to compare two chips in parallel vs two chips in series. 

Author Response

The point-by-pont response to the review comments are attached.

Reviewer 2 Report

Authors proposed a new modular design to generate targeted temporally varying concentration signals in microfluidic systems while minimizing perturbations to the flow field. The work is novel, but I have the following comments:

  1. The quality of figures need to be enhanced
  2. the work lake an in-depth discussion and analysis of the results.

Author Response

(The authors gave the same response as above.)

Reviewer 3 Report

In this paper, a new modular design (referred as LEGO-fluidics) was proposed to generate targeted temporally varying concentration signals in microfluidic systems. The LEGO-fluidics was constructed from a combination of two building blocks which were an oscillator and an integrator. The LEGO-fluidics enabled the creation of controlled and complex concentration signals with different user-defined time-scales. This work is interesting and offers a versatile and promising platform to create highly customizable time-dependent concentration inputs which can be integrated into comprehensive microfluidic systems with specific functions. Before considering for publication, the authors need to make a minor revision on current manuscript.

  1. Real solutions are suggested to be tested on this LEGO-fluidics to characterize the time-dependent concentration inputs. The solution sample can be cell culture medium, drug solution and other common experimental solutions.
  2. Is the modular designing in this work unique? If there are some comparative works, the comparison is suggested to be given.
  3. In the section of conclusion and discussion, authors are suggested to point out the future trend for developing the LEGO-fluidics.

Author Response

(The authors gave the same response as above.)
